# Conditional Mutual Information for Disentangled Representations in Reinforcement Learning

**Mhairi Dunion**
University of Edinburgh
mhairi.dunion@ed.ac.uk

**Trevor McInroe**
University of Edinburgh
t.mcinroe@ed.ac.uk

**Kevin Sebastian Luck**
Vrije Universiteit Amsterdam
k.s.luck@vu.nl

**Josiah P. Hanna**
University of Wisconsin – Madison
jphanna@cs.wisc.edu

**Stefano V. Albrecht**
University of Edinburgh
s.albrecht@ed.ac.uk

## Abstract

Reinforcement Learning (RL) environments can produce training data with spurious correlations between features due to the amount of training data or its limited feature coverage. This can lead to RL agents encoding these misleading correlations in their latent representation, preventing the agent from generalising if the correlation changes within the environment or when deployed in the real world. Disentangled representations can improve robustness, but existing disentanglement techniques that minimise mutual information between features require independent features, thus they cannot disentangle *correlated* features. We propose an auxiliary task for RL algorithms that learns a disentangled representation of high-dimensional observations with correlated features by minimising the *conditional* mutual information between features in the representation. We demonstrate experimentally, using continuous control tasks, that our approach improves generalisation under correlation shifts, as well as improving the training performance of RL algorithms in the presence of correlated features.

## 1 Introduction

Real-world environments are diverse and unpredictable; we often cannot control correlations between environment features or even know they exist. As such, real-world Reinforcement Learning (RL) training environments can contain spurious correlations between features that are unknown or unintended by the data collector, e.g. an object correlated with colour. Furthermore, an RL agent influences the data collection through its actions, which may shift the data distribution to contain feature correlations as the agent learns, e.g. the agent position is correlated with a goal position as it learns an optimal policy. For RL agents to be resilient in the real world, it is beneficial to learn robust representations of high-dimensional observations (e.g. images). However, an agent trained with correlated data may learn a representation that encodes the spurious correlation and, therefore, cannot generalise when the correlation no longer holds (Träuble et al., 2021). For example, an autonomous driving agent trained in an environment where aggressive drivers often have green cars can encode this correlation that does not hold in real world.

Methods for disentanglement aim to separate the ground truth factors of variation that generated a high-dimensional observation, such as an image, into meaningful subspaces in the learned representation (Bengio et al., 2013). Both Higgins et al. (2017b) and Dunion et al. (2023) show that disentanglement improves generalisation to visual changes in RL environments that were not seen during training. A disentangled representation can be robust to environment changes because a change in one image factor causes only a subset of features in the representation to change, while the remaining features can

still be relied upon for better generalisation. The basic principle of most disentanglement techniques is to enforce independence between groups of features in the latent representation. Therefore, common approaches to disentanglement, such as VAEs, require independence between factors of variation, i.e. uncorrelated factors. RL agents trained to learn independent features in the representation will fail to separate correlated factors because they cannot be separated into distinct independent features, and thus suffer from a failure to generalise under correlation shifts. For example, consider a scenario where the colour of an object is correlated with its size during training, and size impacts the optimal policy. Colour contains information predictive of size and vice versa. Hence, size and colour cannot be separated into distinct *independent* features in the representation, so will be encoded into the same feature. When the agent is presented with a colour that is rarely or never previously seen with the object size, the feature representing both colour and size will change, preventing the agent from performing optimally even if it has already learned an optimal policy for the object size.

In this work, we relax the assumption of independence between factors of variation to *conditional* independence, to learn a disentangled representation with correlated features. We propose Conditional Mutual Information for Disentanglement (CMID) as an auxiliary task that can be applied to online RL algorithms to learn a disentangled representation by minimising the conditional mutual information between dimensions in the latent representation. We use the causal graph of a Markov Decision Process (MDP) to determine a general conditioning set that can render the features in a representation conditionally independent given the conditioning set. The resulting disentangled representation avoids the reliance on spurious correlations, separating correlated factors of variation in a high-dimensional observation into distinct features in the representation, allowing for better generalisation under correlation shifts. To the best of our knowledge, this is the first approach to learn disentangled representations specifically for RL based on *conditional* independence.

We evaluate our approach on continuous control tasks with image observations from the DeepMind Control Suite where we add correlations between object colour and properties impacting dynamics (e.g. joint positions). Our results show that CMID improves the training performance as well as the generalisation performance of the base RL algorithm, SVEA (Hansen et al., 2021), under correlation shifts, with a 77% increase in zero-shot generalisation returns on average across all tasks in our experiments. We also demonstrate improved generalisation performance compared to state-of-the art baselines: DrQ (Yarats et al., 2021), CURL (Laskin et al., 2020b) and TED (Dunion et al., 2023).

## 2 Related work

### 2.1 Disentangled representations

**VAE approaches.** Disentanglement has been studied widely in the unsupervised learning literature. Many approaches are based on the Variational Autoencoder (VAE) (Kingma and Welling, 2014). The $\beta$-VAE (Higgins et al., 2017a; Burgess et al., 2017) aims to improve disentanglement by scaling up the independence constraint in the VAE loss; the Factor-VAE (Kim and Mnih, 2018) encourages disentanglement through a factorial distribution of features. More recent approaches add supervision during training to bypass the impossibility of learning disentangled representations from independent and identically distributed (i.i.d.) data (Locatello et al., 2019). Shu et al. (2020) use labels of image groupings, and Locatello et al. (2020) use pairs of images. However, VAE-based approaches assume independence between factors of variation and therefore cannot disentangle correlated factors.

**ICA approaches.** Disentanglement is also the focus of Independent Component Analysis (ICA) (Hälvä et al., 2021; Hyvärinen et al., 2023), where it is referred to as 'blind source separation'. Hyvärinen and Morioka (2016) and Hyvärinen and Morioka (2017) both learn disentangled representations from time-series data under different assumptions. However, similarly to VAE approaches, the independence assumption is central to ICA and does not hold when there are correlations in the data.

**Disentanglement with correlated features.** Both ICA and VAE approaches to disentanglement assume independence between factors of variation (or 'sources' in ICA terminology). Träuble et al. (2021) conduct an analysis of VAE-based disentanglement techniques on correlated data and show that the correlations are being encoded in the representations. They propose weak supervision in training to learn disentangled representations on correlated data and an adaptation technique to 'correct' the latent entanglement using labeled data. Recently, Funke et al. (2022) propose a Conditional Mutual

Information (CMI) approach to learn disentangled representations to improve generalisation under correlation shifts in a supervised learning setting. We use a similar adversarial approach to CMI as Funke et al. (2022), but leverage the structure of an MDP to determine an appropriate conditioning set that does not require labelled data or any prior knowledge of the ground truth factors of variation.

## 2.2 Representation learning in RL

**Visual invariances.** Image augmentations are commonly used to improve robustness of representations in RL (Laskin et al., 2020a; Yarats et al., 2021; Hansen and Wang, 2021; Hansen et al., 2021). Several approaches have also been proposed to learn representations that are invariant to distractors in the image such as background colour (Zhang et al., 2020, 2021; Li et al., 2021; Allen et al., 2021). These methods do not account for correlations between features and do not prevent the encoding of spurious correlations. Mutual information has recently been used for invariant representation learning outside of RL by Cerrato et al. (2023) to ensure model decisions are independent of specific input features.

**Mutual information.** Mutual information (MI) based approaches are commonly used in RL for representation learning. Laskin et al. (2020b) maximise similarity between different augmentations of the same observation; Mazoure et al. (2020) maximise similarity between successive observations; and Agarwal et al. (2021) use policy similarity metrics. However, these approaches all maximise MI in some way, whereas disentanglement aims to minimise MI between features in the representation.

**Disentangled representations.** To learn a disentangled representation for RL, Higgins et al. (2017b) train a $\beta$-VAE offline using i.i.d. data from a pre-trained agent. Dunion et al. (2023) propose an auxiliary task to learn disentangled representations online using the non-i.i.d. temporal structure of RL training data. Both of these approaches to disentanglement in RL assume independent factors of variation, so they are unable to disentangle correlated factors.

## 2.3 Conditional mutual information estimators

Our approach is the first to use CMI for disentangled representations in RL. However, many approaches have been proposed to estimate CMI outside of RL. Initial approaches were extensions of MI estimators, such as Runge (2018). Recent approaches make use of advances in neural networks. Mukherjee et al. (2020) propose CCMI, using the difference between two MI terms for CMI estimation: $I(X;Y \mid Z) = I(X;Y,Z) - I(X;Z)$. They propose an estimator for the KL-divergence by training a classifier to distinguish the observed joint distribution from the product distribution. Mondal et al. (2020) estimates CMI by re-formulating it as a minmax optimisation problem and using a training procedure similar to generative adversarial networks. Molavipour et al. (2021) extend the classifier approach of CCMI, applying it directly to the estimation of CMI, rather than the difference of two MI terms. They use $k$ nearest neighbours (kNN) to sample from the product of marginals distribution, and train the classifier to distinguish between the original distribution and the product of marginals. We also use kNN permutations to sample from the product of marginals distribution.

## 3 Preliminaries

**Reinforcement learning.** We assume the agent is acting in a Markov Decision Process (MDP), which is defined by the tuple $\mathcal{M} = (\mathcal{S}, \mathcal{A}, P, R, \gamma)$, where $\mathcal{S}$ is the state space, $\mathcal{A}$ is the action space, $P(s_{t+1}|s_t, a_t)$ is the probability of next state $s_{t+1}$ given action $a_t \in \mathcal{A}$ is taken in state $s_t \in \mathcal{S}$ at time $t$, $R(s_t, a_t)$ is the reward function, and $\gamma \in [0, 1)$ is the discount factor. The goal is to learn a policy $\pi$ that maximises the discounted return, $\max_\pi \mathbb{E}_{P,\pi}[\sum_{t=0}^\infty [\gamma^t R(s_t, a_t)]]$. In RL from pixels, the agent receives an observation of image pixels $\mathbf{o}_t \in \mathcal{O} \subset \mathbb{R}^{i \times j}$ at time $t$, a high-dimensional representation of $s_t$. The agent learns a latent representation $\mathbf{z}_t = f_\theta(\mathbf{o}_t)$ of size $N \ll \dim(\mathcal{O})$, where $f_\theta : \mathcal{O} \to \mathcal{Z}$ is an encoder parameterised by $\theta$. The policy $\pi$ is a function of the latent representation, such that $\mathbf{a}_t \sim \pi(\mathbf{z}_t)$. We denote the $n$-th component of the vector $\mathbf{z}$ as $z^n$, and all components of $\mathbf{z}$ except $z^n$ as $\mathbf{z}^{-n}$. We use $\mathbf{z}_{t':t''}$ to refer to representations for all consecutive timesteps from $t'$ to $t''$ inclusive: $\mathbf{z}_{t'}, \mathbf{z}_{t'+1}, ..., \mathbf{z}_{t''-1}, \mathbf{z}_{t''}$.

**Conditional mutual information.** The Conditional Mutual Information (CMI) of continuous random variables $X$ and $Y$ given a third variable $Z$ measures the amount of information $Y$ contains

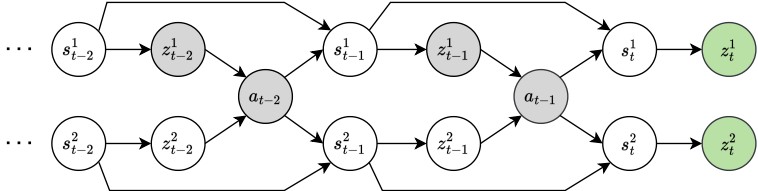

**Figure 1:** The conditioning set for an MDP is highlighted in grey. Representation features $z_t^1$ and $z_t^2$ are conditionally independent when conditioned on $\mathbf{z}_{0:t-1}^1$ and $\mathbf{a}_{0:t-1}$, where $z_t^n$ denotes the $n$th dimension of $\mathbf{z}_t$.

about $X$ given $Z$ is already known (Cover and Thomas, 2006), defined as:

$$I(X;Y \mid Z) := \iiint p(x,y,z) \log \frac{p(x,y,z)}{p(x,z)p(y|z)} dxdydz \qquad (1)$$

where lower-case letters denote instances of a random variables (e.g. $x$ is an instance of $X$). By definition, CMI is given by the KL-divergence:

$$I(X;Y \mid Z) = D_{\mathrm{KL}}[p(x,y,z) \mid\mid p(x,z)p(y|z)]. \qquad (2)$$

If the CMI between $X$ and $Y$ given $Z$ is 0, then $X$ and $Y$ are conditionally independent given $Z$:

$$I(X;Y \mid Z) = 0 \iff X \perp\!\!\!\perp Y \mid Z. \qquad (3)$$

## 4   Conditional mutual information for disentanglement in RL

We propose Conditional Mutual Information for Disentanglement (CMID) as an auxiliary task that can be applied to existing RL algorithms to learn a disentangled representation with correlated data. The goal is to learn a representation with features that are conditionally independent to improve representation robustness in the presence of unintended correlations during training. We discuss the conditioning set for an MDP in Section 4.1 and describe the CMID auxiliary task in Section 4.2.

### 4.1   MDP conditioning set

The causal graph for three timesteps of an MDP is shown in Figure 1. For readability, the graph shows only two state features $s^1$ and $s^2$, and two representation features $z^1$ and $z^2$, however the graph and subsequent discussion can be extended to an arbitrary number of features. The graph shows the desired causal relationships for the learned representation, such that each feature in the representation is caused by a single state feature. An overview of the relevant concepts from causality that we will use in this section is provided in Appendix A.2.

The goal is to learn a representation $\mathbf{z}_t$ where features $z_t^1$ and $z_t^2$ are conditionally independent by blocking all backdoor paths between $z_t^1$ and $z_t^2$ in the causal graph (Pearl, 2009). One option is to condition on the true underlying state feature $s_t^1$, which is the approach taken by Funke et al. (2022), but we do not usually know the true state features. Given the temporal structure of an MDP, another suitable conditioning set is the history of $z_t^1$, denoted $\mathbf{z}_{0:t-1}^1$, and the history of actions $\mathbf{a}_{0:t-1}$, giving:

$$z_t^1 \perp\!\!\!\perp z_t^2 \mid \mathbf{z}_{0:t-1}^1, \mathbf{a}_{0:t-1}. \qquad (4)$$

In other words, $z_t^2$ does not contain any additional information about $z_t^1$ given $\mathbf{z}_{0:t-1}^1$ and $\mathbf{a}_{0:t-1}$ are known. Similarly, the history of $z_t^2$ would also make a suitable conditioning set, since the backdoor path can be blocked by $\mathbf{a}_{0:t-1}$ and either $\mathbf{z}_{0:t-1}^1$ or $\mathbf{z}_{0:t-1}^2$. Conditioning on the history of actions $\mathbf{a}_{0:t-1}$ alone is not sufficient because the action is a collider in the causal graph, so the conditioning set must also contain a parent of this collider to avoid opening up new backdoor paths (Pearl, 2009). This conditioning set means that we do not need to know the true state features $\mathbf{s}_t$ to learn conditionally independent representation features $z_t^1$ and $z_t^2$.

To guarantee conditional independence, the conditioning set must include the full history $\mathbf{z}_{0:t-1}^1$ and $\mathbf{a}_{0:t-1}$ from the beginning of the episode to the previous timestep $t-1$. However, conditioning on the full history makes a very large conditioning set, of size $t \cdot (N + \dim(\mathbf{a}))$ when using one-hot

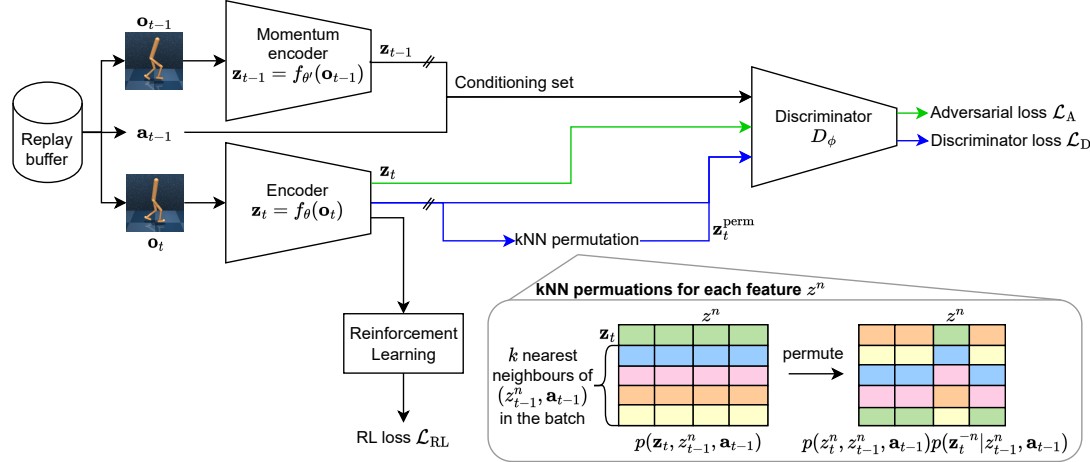

**Figure 2:** CMID architecture: The discriminator learns to discriminate between samples from $p(\mathbf{z}_t \mid z_{t-1}^n, \mathbf{a}_{t-1})$ and $p(z_t^n \mid z_{t-1}^n, \mathbf{a}_{t-1})p(\mathbf{z}_t^{-n} \mid z_{t-1}^n, \mathbf{a}_{t-1})$. The encoder is trained adversarially to make the two distributions similar to minimise CMI. The double slash '//' on the encoder outputs indicates where gradient flow is stopped.

encoding of features, which can be difficult to learn. In practice, we condition on only the most recent timestep $z_{t-1}^1$ and $\mathbf{a}_{t-1}$ to adjust for the most recent correlations while keeping the auxiliary task reasonable to achieve during training. We show experimentally in Section 6 that conditioning on only the most recent timestep achieves good generalisation performance while converging to an optimal policy faster in training than larger conditioning sets that use more timesteps from the episode history.

### 4.2 Conditional Mutual Information for Disentanglement

CMID is an auxiliary task to learn disentangled representations. The architecture for CMID is shown in Figure 2, and the pseudocode is provided in Algorithm 1. CMID uses the same image inputs as the base RL algorithm. Where the base algorithm uses image augmentations (Hansen et al., 2021; Yarats et al., 2021; Laskin et al., 2020a), these are also used for CMID. However, CMID does not use frame stacking because it would introduce causal relationships between features in the frame stack. For example, object velocity and positions are often extracted from the frame stack, but velocity is also a direct cause of position. CMID processes each frame individually, and representations can be stacked if required to allow velocity information to be extracted by the RL networks for policy learning.

To learn a conditionally independent representation, CMID minimises the CMI between features in the representation. For each feature $z_t^n$ in the representation $\mathbf{z}_t$, it follows from Equation 2 and the conditioning set $\mathbf{c}_t^n = (z_{t-1}^n, \mathbf{a}_{t-1})$ discussed in Section 4.1 that:

$$I(z_t^n; \mathbf{z}_t^{-n} \mid \mathbf{c}_t^n) = D_{\text{KL}}\left[p(\mathbf{z}_t, \mathbf{c}_t^n) \mid\mid p(z_t^n, \mathbf{c}_t^n)p(\mathbf{z}_t^{-n} \mid \mathbf{c}_t^n)\right]. \tag{5}$$

As such, we minimise the KL-divergence between the joint probability distribution $p(\mathbf{z}_t, \mathbf{c}_t^n)$ and the product of marginals $p(z_t^n, \mathbf{c}_t^n)p(\mathbf{z}_t^{-n} \mid \mathbf{c}_t^n)$. The agent has access to samples from the joint distribution $p(\mathbf{z}_t, \mathbf{c}_t^n)$ collected during training, e.g. from the replay buffer. To sample from the product of marginals, we use the isolated $k$ nearest neighbours (kNN) permutation approach (Molavipour et al., 2021). For each sample $\{\mathbf{z}_t, \mathbf{c}_t^n\} \sim p(\mathbf{z}_t, \mathbf{c}_t^n)$, we find the kNN of $\mathbf{c}_t^n$ by Euclidean distance, then permute the sample with the kNN to get a sample $\{z_t^n, \mathbf{z}_{t'}^{-n}, \mathbf{c}_t^n\}$ where $t' \neq t$ and $\mathbf{c}_{t'}^n$ (the conditioning set of $z_{t'}$ that is used for the permutation) is a kNN of $\mathbf{c}_t^n$. We will use $\mathbf{z}_t^{\text{perm},n}$ to denote the permutations $\{z_t^n, \mathbf{z}_{t'}^{-n}\}$. The permuted sample $\{\mathbf{z}_t^{\text{perm},n}, \mathbf{c}_t^n\}$ is from the distribution $p(z_t^n, \mathbf{c}_t^n)p(\mathbf{z}_t^{-n} \mid \mathbf{c}_t^n)$. The permutation process is also depicted in Figure 2.

To minimise the KL-divergence in Equation 5, we train a discriminator $D_\phi$ adversarially to distinguish between samples $\{\mathbf{z}_t, \mathbf{c}_t^n\} \sim p(\mathbf{z}_t, \mathbf{c}_t^n)$ and $\{\mathbf{z}_t^{\text{perm},n}, \mathbf{c}_t^n\} \sim p(z_t^n, \mathbf{c}_t^n)p(\mathbf{z}_t^{-n} \mid \mathbf{c}_t^n)$. The adversarial training encourages the encoder $f_\theta$ to ensure the two distributions are as similar as possible by minimising the cross entropy. This objective is equivalent to minimising the KL-divergence since for any two distributions $p$ and $q$, $D_{\text{KL}}(p \mid\mid q) = H(p, q) - H(p)$ where $H(p, q)$ is the cross entropy and $H(p)$ is the entropy of $p$ which does not depend on the learned parameters. The discriminator

**Algorithm 1** CMID update step

---

**Input:** batch of transitions $B = \{..., (\mathbf{o}_{t-1}, \mathbf{a}_{t-1}, \mathbf{o}_t), ...\} \sim \mathcal{D}$
**Input:** parameters for the encoder $\theta$ and the discriminator $\phi$
Calculate RL loss $\mathcal{L}_{\text{RL}}$ and update RL networks (including encoder) following base RL algorithm
Initialise $\mathcal{L}_{\text{D}} \leftarrow 0$ and $\mathcal{L}_{\text{A}} \leftarrow 0$
Forward pass though encoder $\mathbf{z}_t = f_\theta(\mathbf{o}_t)$ and momentum encoder $\mathbf{z}_{t-1} = f_{\theta'}(\mathbf{o}_{t-1})$
**for** $n \in (1, ..., N)$ **do**
    Create conditioning set $\mathbf{c}_t^n = (z_{t-1}^n, \mathbf{a}_{t-1})$
    Find $k$ nearest neighbours of $\mathbf{c}_t^n$ in the batch, measured by the Euclidean distance: $\sqrt{\sum_i ((\mathbf{c}_t^n)^i - (\mathbf{c}_{t'}^n)^i)^2}$
    Create permuted samples $\mathbf{z}_t^{\text{perm},n}$ by shuffling $k$ nearest neighbours, keeping $z_t^n$ fixed
    Calculate discriminator loss $\mathcal{L}_{\text{D}} \leftarrow \mathcal{L}_{\text{D}} + \log \sigma(D_\phi(\mathbf{z}_t, \mathbf{c}_t^n)) + \log(1 - \sigma(D_\phi(\mathbf{z}_t^{\text{perm},n}, \mathbf{c}_t^n)))$
**end for**
Update discriminator parameters to minimise $\mathcal{L}_{\text{D}}$
**for** $n \in (1, ..., N)$ **do**
    Create conditioning set $\mathbf{c}_t^n = (z_{t-1}^n, \mathbf{a}_{t-1})$
    Calculate adversarial loss $\mathcal{L}_{\text{A}} \leftarrow \mathcal{L}_{\text{A}} + \log(1 - \sigma(D_\phi(\mathbf{z}_t, \mathbf{c}_t^n)))$
**end for**
Update encoder parameters to minimise $\mathcal{L}_{\text{A}}$
**Output:** Losses $\mathcal{L}_{\text{D}}, \mathcal{L}_{\text{A}}, \mathcal{L}_{\text{RL}}$ and updated parameters $\phi, \theta$

---

$D_\phi$ is trained to discriminate between true and permuted samples for each feature $z_t^n$ using a binary cross entropy loss:

$$\mathcal{L}_{\text{D}} = \frac{1}{N} \sum_{i=0}^{N} \left( \log \sigma(D_\phi(\mathbf{z}_t, \mathbf{c}_t^n)) + \log(1 - \sigma(D_\phi(\mathbf{z}_t^{\text{perm},n}, \mathbf{c}_t^n))) \right) \tag{6}$$

where $\sigma$ is the sigmoid function. The encoder $f_\theta$ is updated using only true samples $\mathbf{z}_t$ to learn a representation such that the discriminator cannot determine whether the sample is true or permuted:

$$\mathcal{L}_{\text{A}} = \frac{\alpha}{N} \sum_{i=0}^{N} \log(1 - \sigma(D_\phi(\mathbf{z}_t, \mathbf{c}_t^n))). \tag{7}$$

The loss coefficient $\alpha$ is a hyperparameter to be tuned to the task to determine the scale of the adversarial loss compared to the RL loss that also updates the encoder $f_\theta$. For training stability, we use a momentum encoder (He et al., 2020; Laskin et al., 2020b) $f_{\theta'} : \mathcal{O} \rightarrow \mathcal{Z}$ for the conditioning set representation $\mathbf{z}_{t-1} = f_{\theta'}(\mathbf{o}_{t-1})$, where $\theta' = \tau\theta' + (1 - \tau)\theta$.

## 5 Experimental results

Our experiments evaluate generalisation performance under correlation shifts. We evaluate zero-shot generalisation as well as adaptation with continued learning on test correlations (which differ from the training correlations). We test our approach on continuous control tasks from the DeepMind Control Suite (DMC) (Tunyasuvunakool et al., 2020) where we have created strong correlations between task-relevant features and irrelevant colours. We use a training environment with correlated variables and evaluate generalisation on a test environment under correlation shift. Our results show that the CMID auxiliary task consistently improves the generalisation of the base RL algorithm in all tasks, as well as outperforming other baselines.

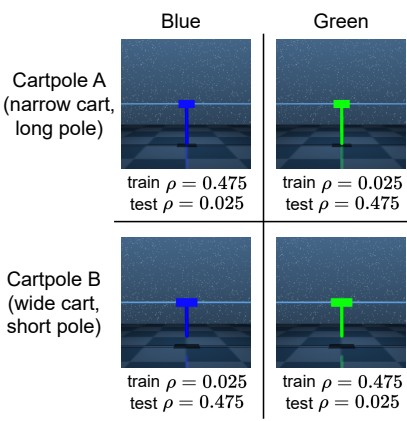

**Figure 3:** Illustration of correlations with testing on reversed correlation in the cartpole environment, $\rho$ indicates the probability of an object/colour combination.

### 5.1 Experimental setup

**Adding strong correlations to DMC.** To demonstrate generalisation under correlation shifts, we add correlations between object colour and dynamics. We use two

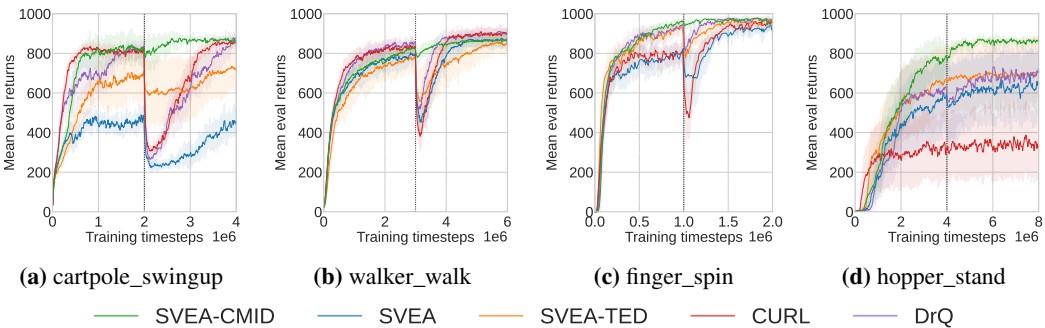

**Figure 4:** Generalisation to *reversed correlation* at the vertical dotted line. Returns are the average of 10 evaluation episodes, averaged over 5 seeds; the shaded region is the standard error.

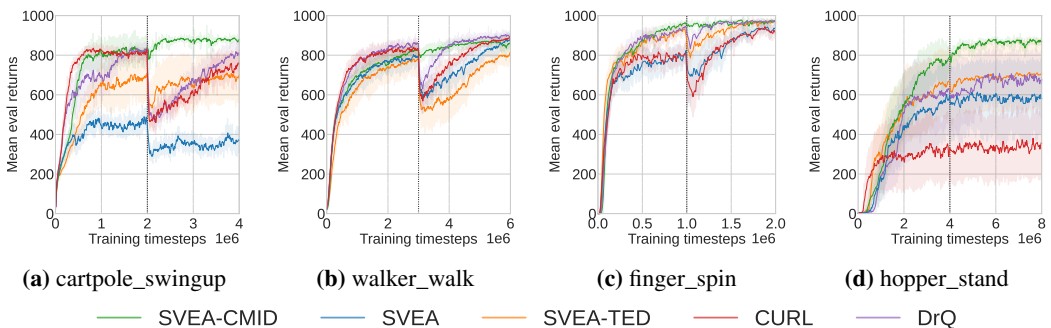

**Figure 5:** Generalisation to *uncorrelated* features at the vertical dotted line. Returns are the average of 10 evaluation episodes, averaged over 5 seeds; the shaded region is the standard error.

variations of the object controlled by the agent (A and B), each of which has a slightly different morphology (e.g. lengths, joint positions), which affects both the object appearance and the dynamics. This means each variation of the control object requires a different optimal policy. A description of the morphology variations for each task along with images are provided in Appendix D. At the start of an episode, object A or B is chosen at random with equal probability. Object A appears blue with probability 0.95 and green with probability 0.05, conversely object B is green with probability 0.95 and blue otherwise. After training, the correlation is changed (at the vertical dotted line in the graphs) and the agent continues training to assess adaptation. We test two different correlation shifts: reversed correlation, and no correlation (i.e. each object is equally likely to be blue or green). An example of the correlated setup with testing on reversed correlation for cartpole is shown in Figure 3, and these correlation probabilities are used across all DMC tasks unless stated otherwise.

**Base RL algorithm.** We use SVEA (Hansen et al., 2021) as the base RL algorithm which we augment with the CMID auxiliary task, called SVEA-CMID in the results. SVEA is used because it is a state-of-the-art RL algorithm that already uses image augmentations to improve robustness to an extent. An overview of SVEA is provided in Appendix A.1 and implementation details in Appendix B.

**Baselines.** We compare with SVEA, as the base RL algorithm for CMID, to demonstrate the extent to which the CMID auxiliary task improves performance. We also compare with DrQ (Yarats et al., 2021) as an alternative image augmentation baseline. We compare with CURL (Laskin et al., 2020b) as a state-of-the-art auxiliary task based on maximising mutual information. As a disentanglement baseline that assumes independent features, we use TED (Dunion et al., 2023) to demonstrate that disentanglement techniques requiring fully independent features fail with strong correlations.

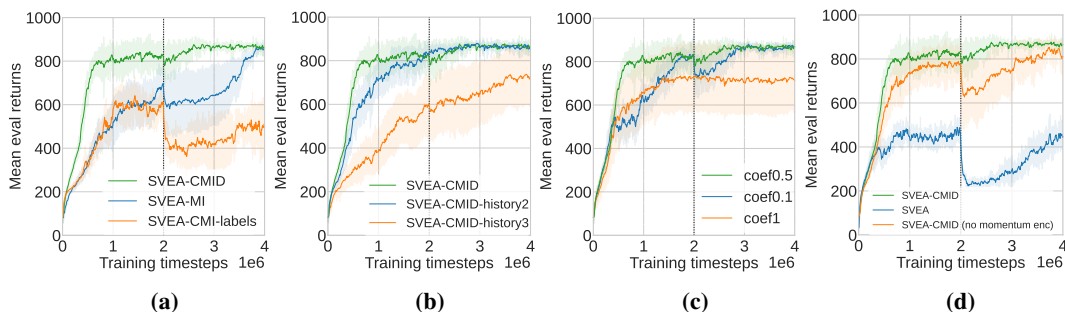

**Figure 6:** Ablation experiments on cartpole with generalisation to reversed correlation at the vertical dotted line: **(a)** comparison with MI and labelled features, **(b)** varying history lengths in the conditioning set, **(c)** different values of the CMID coefficient $\alpha$, and **(d)** CMID with and and without the momentum encoder.

## 5.2 Generalisation results

Our results in Figure 4 show the generalisation to reversed correlation and Figure 5 shows generalisation to uncorrelated features at the vertical dotted line. In both cases, the results show that CMID improves the generalisation performance of SVEA in all tasks, as well as outperforming the other baselines. CMID achieves good zero-shot generalisation performance on the vertical dotted line, while all baselines have some failure to generalise (except where they cannot learn a reasonable policy in the hopper task). Tables showing the numerical values of the zero-shot generalisation performance are also provided in Appendix C.1. Some baselines are able to to adapt with continued learning on the test environment to eventually achieve optimal performance in line with CMID, but others are unable to recover an optimal policy after overfitting to the training correlations. CMID also improves the training performance of SVEA in all tasks, achieving higher training returns even before the switch to the test environment. Many baselines also suffer from this inability to achieve optimal performance in training because the strong correlation makes it harder to learn as an optimal agent needs to learn a different policy for each control object without relying on the colour to distinguish between control objects. Appendix C.2 shows the evaluation performance on each scenario for cartpole swingup to further demonstrate why the failure to learn an optimal policy and generalise occurs.

## 6 Discussion and analysis

In this section, we conduct more detailed analysis of CMID on the reverse correlation testing scenario for the cartpole swingup task from DMC. We also provide some further analysis on correlation strength and greyscale images in Appendix C.3 and C.4 respectively.

**Mutual information.** To validate that it is necessary to minimise the *conditional* MI in our approach, we compare with an (unconditional) MI variant with no conditioning set. The results in Figure 6a show that SVEA-CMID outperforms SVEA-MI in training performance and generalisation. To validate our conditioning set, we also compare to the CMI approach of Funke et al. (2022) modified to an RL setting. This approach, which we call SVEA-CMI-labels, assumes access to the ground truth state features. The representation is split into subspaces corresponding to each of the state features, and a classifier is trained for each subspace to predict the state feature. The true state features can then be used as the conditioning set. Figure 6a also shows that SVEA-CMID outperforms SVEA-CMI-labels because of the added complexity of using the true state features and training the classifiers.

**History length.** We discussed in Section 4.1 that achieving full conditional independence requires conditioning on the history of representations $\mathbf{z}_{0:t-1}$ and actions $\mathbf{a}_{0:t-1}$. However, in practice (Section 5) we found that conditioning only on the most recent representation $\mathbf{z}_{t-1}$ and action $\mathbf{a}_{t-1}$ breaks the strongest correlation to achieve zero-shot generalisation. Figure 6b compares CMID with variations that condition on the previous two (CMID-history2) and three (CMID-history3) timesteps. The results show that it is harder to learn with the larger conditioning sets, which fail to achieve optimal performance as quickly, while CMID with one previous timestep works well in practice.

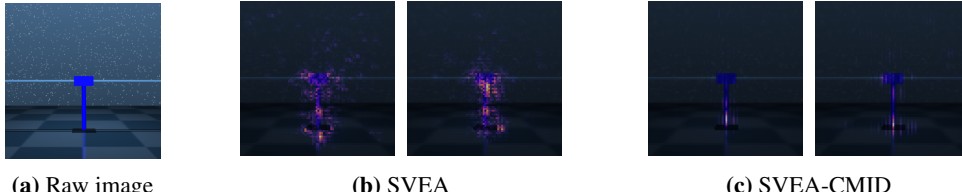

| **(a)** Raw image | **(b)** SVEA | **(c)** SVEA-CMID |

**Figure 7:** Saliency maps: (a) raw image used to calculate attributions, (b) SVEA and (c) SVEA-CMID saliency maps showing two representation features. Brighter pixels correspond to higher attributions. SVEA-CMID has designated features focusing on the pole length which it has disentangled from other features.

**CMID loss coefficient.** The CMID loss coefficient $\alpha$ in Equation 7 is a hyperparameter to be tuned to the task. We found $\alpha = 0.5$ performs well in the cartpole task. The results in Figure 6c show that decreasing $\alpha$ decreases generalisation performance because the agent is not prioritising disentanglement, as well as reducing training performance because it is harder to learn the rare cases with this lower priority on disentanglement. Increasing $\alpha$ makes it harder to learn the optimal policy, decreasing the training performance.

**Momentum encoder.** The momentum encoder is not a strictly necessary component for minimising CMI, but we found empirically that it improves stability as it does for some other RL algorithms (He et al., 2020; Laskin et al., 2020b). All algorithms used in our experiments are implemented with a momentum encoder, so we make use of the momentum encoder that is already available for many algorithms. Figure 6d shows the results for SVEA-CMID on the cartpole swingup task with and without a momentum encoder. CMID without a momentum encoder still improves the performance of SVEA but does not perform as well as the momentum encoder version and has higher variance.

**Visualising the learned representation.** Existing disentanglement metrics assume independent factors of variation so are not suitable to measure disentanglement on correlated data (Träuble et al., 2021). Instead we conducted qualitative analysis of the learned representation to visualise the disentanglement. We use integrated gradients (Sundararajan et al., 2017) to attribute the encoder output value of each feature in the representation to the input image pixels. We overlay the attributions on the original image to create saliency maps showing the parts of the image that each representation feature focuses on. We show illustrative saliency maps in Figure 7. Implementation details and saliency maps for all representation features are provided in Appendix E. The saliency maps show that SVEA encodes many image features in one representation feature, while SVEA-CMID has designated features in the representation to focus on individual features in the image, such as pole length, which is necessary to distinguish between cartpole A and B.

**Robustness analysis.** We analyse the robustness of the trained RL agents to unseen colours on the cartpole swingup task. Using the model at the end of training (before changing the correlation), we test the model on the same control objects but with unseen colours. We test on 216 different colours, using equally spaced RGB values. The results in Table 1 show that CMID achieves improved zero-shot generalisation performance on the unseen colours, in terms of the worst performing colour, the best performing colour and the average.

|  | SVEA | SVEA-CMID |
|---|---|---|
| Worst colour | $165.8 \pm 13.8$ | $\mathbf{379.1 \pm 70.1}$ |
| Best colour | $588.7 \pm 87.2$ | $\mathbf{834.1 \pm 105.8}$ |
| Average | $220.2 \pm 27.4$ | $\mathbf{692.2 \pm 166.3}$ |

**Table 1:** Zero-shot generalisation to unseen colours on the cartpole task. Mean returns on 10 evaluation episodes over 5 seeds.

## 7    Limitations and future work

Instead of the common practice of frame stacking, we stack representations when using CMID to avoid introducing causal relationship between variables in the stack of frames as discussed in Section 4.2. Future work could consider how to adapt CMID to allow for these more complex causal relationships. The kNN permutations approach to minimise CMI also adds computational complexity to update the encoder for each kNN, adding a 67% increase in run time on average for our experiments

compared to the base RL algorithm. The number of kNN and the size of the representation could be reduced in scenarios where computation time is of high importance, and future work could consider more efficient ways to sample from the product of marginals distribution.

We evaluated our approach on tasks with correlation between object properties (that impact dynamics) and colour. This scenario already shows that state-of-the-art baselines suffer from a significant deterioration in performance under correlation shifts as well as being unable to learn an optimal policy in training for some tasks. As such, the colour correlations are sufficient to demonstrate the effectiveness of our approach in improving generalisation. However, future work could evaluate our approach on correlations with more complex distractors, such as background videos and camera angles. In particular, we use a conditioning set containing only the most recent timestep in our experiments, but more complex environments can have strong correlations over multiple timesteps (e.g. background videos) which may require more history in the conditioning set (Albrecht and Ramamoorthy, 2016). Future work could consider using more representations and actions from history in the conditioning set efficiently to apply CMID to more complex environments and correlations.

Finally, CMID learns a disentangled representation while exploring using the same strategy as used by the base RL algorithm. However, it is possible that the learning agent could discover a disentangled representation faster through a new exploration strategy that actively probes the environment to determine state structure. In the future, we plan to investigate the combination of CMID with recent advances in exploration (Henaff, 2019; Sontakke et al., 2021; Schäfer et al., 2022; Zhong et al., 2022; McInroe et al., 2023) to see whether these advances allow the CMID agent to more quickly discover disentangled representations.

## 8 Conclusion

In this paper, we explored the problem of training with strong correlations and generalisation under correlation shifts in RL. Existing techniques for learning disentangled representations in RL are insufficient for real-world problems because they assume the ground truth features are independent, which is unlikely to hold in practice. We proposed Conditional Mutual Information for Disentanglement (CMID) to learn disentangled representations with correlated image features that requires only conditional independence between features. CMID is an auxiliary task that can be used with existing RL algorithms. We showed experimentally that CMID improves the training and generalisation performance of SVEA as the base RL algorithm as well as DrQ, CURL and TED baselines. CMID allows the RL agent to generalise under correlation shifts and continue learning without performance reduction as a step towards training on real-world correlated data.

## 9 Acknowledgements

This work was supported by the United Kingdom Research and Innovation (grant EP/L016834/1), EPSRC Centre for Doctoral Training in Robotics and Autonomous Systems (RAS) in Edinburgh. This work was also supported by the Academy of Finland Flagship programme: Finnish Center for Artificial Intelligence FCAI. The authors wish to acknowledge the generous computational resources provided by the Aalto Science-IT project and the CSC – IT Center for Science, Finland.

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

# A  Extended background

## A.1  Reinforcement Learning

We use SVEA (Hansen et al., 2021) as the base RL algorithm for the CMID auxiliary task, which is an extension of the Soft Actor-Critic (SAC) algorithm (Haarnoja et al., 2018).

SAC is an off-policy RL algorithm for continuous control. SAC learns a stochastic policy $\pi$ that maximises the expected sum of rewards and the entropy of the policy. The critic $Q$ is learned by minimising the loss:

$$L_Q = \mathbb{E}_{(\mathbf{o}_t, \mathbf{a}_t, \mathbf{o}_{t+1}, r_t) \sim \mathcal{D}} \left[ \left( Q(\mathbf{o}_t, \mathbf{a}_t) - r_t - \gamma \bar{V}(\mathbf{o}_{t+1}) \right)^2 \right] \tag{8}$$

where $\mathbf{o}_t$ is the image observation and $\mathbf{a}_t$ is the action at time $t$ as defined in Section 3. SAC uses the minimum of two $Q$ networks, $Q_1$ and $Q_2$, for the training updates to reduce overestimation of $Q$ values. The actor $\pi$ is trained by minimising the loss:

$$L_\pi = \mathbb{E}_{\mathbf{o}_t \sim \mathcal{D}} \left[ \mathbb{E}_{\mathbf{a}_t \sim \pi} \left[ \alpha_{\text{SAC}} \log(\pi(\mathbf{a}_t \mid \mathbf{o}_t)) - \min_{i=1,2} \bar{Q}_i(\mathbf{o}_t, \mathbf{a}_t) \right] \right] \tag{9}$$

where $\bar{Q}$ is exponential moving average of the Q network parameters.

SVEA aims to stabilise SAC training using a combination of both augmented and unaugmented images for $Q$ learning with an modified loss:

$$L_Q^{\text{SVEA}} = \alpha_{\text{SVEA}} L_Q(o_t, a_t, o_{t+1}) + \beta_{\text{SVEA}} L_Q(o_t^{\text{aug}}, a_t, o_{t+1}) \tag{10}$$

However, the actor $\pi$ is optimised on unaugmented images only, using the SAC policy loss in Equation 9.

## A.2  Causality

We will provide a brief overview of the relevant concepts from causal inference used in Section 4.1, and we refer the interested reader to the book by Pearl (2009) for details.

A causal graph is a directed acyclic graph. The nodes in the graph correspond to random variables and the directed edges represent a causal relationship between two variables. The causal graph defines the (in)dependence between the variables. Two variables $X$ and $Y$ can be considered separated by $Z$ in a causal graph if $X$ is independent of $Y$ given the conditioning set $Z$. In other words, once the value of $Z$ is known, knowing the value of $X$ will no longer influence the belief about $Y$. This condition is called *separation* in the graph and forms the link between *blocking paths* in the causal graph and (in)dependencies in the data. A path in the graph is a sequence of consecutive edges, and a *backdoor path* is the non-causal path between $X$ and $Y$ containing no descendants of $X$, i.e. the paths that flow "backwards" from $X$. A path between two variables $X$ and $Y$ is *blocked* by a set of nodes $Z$ (the conditioning set) if the following conditions hold (Pearl, 2009):

1. if the path contains a chain $X \to M \to Y$, then a node in the mediator set $M$ is in $Z$

2. if the path contains a fork $X \leftarrow U \to Y$, then a node in the confounder set $U$ is in $Z$

3. if path contains a collider $X \to C \leftarrow Y$ then the collider node $C$ is *not* in $Z$ and no descendant of $C$ is in $Z$.

A collider node $C$ naturally blocks a path that traces it, so conditioning on a collider (or a descendant of a collider), opens the path. As such, where conditioning on a collider is necessary, then the conditioning set should also include variables that block the newly opened path. If all paths between $X$ and $Y$ are blocked by $Z$ then $X$ is independent of $Y$ given $Z$.

# B  Implementation details

In this section, we provide the implementation details for CMID. Our codebase is built on top of the publicly released DrQ PyTorch implementation by Yarats et al. (2021) as well as the official implementation of SVEA by Hansen et al. (2021). A public and open-source implementation of CMID is available at github.com/uoe-agents/cmid.

| Hyperparameter | Value |
|---|---|
| Replay buffer capacity | 100000 |
| Initial steps before training begins | 1000 |
| Stacked frames (stacked representations for CMID) | 3 |
| Action repeat | 2 for finger_spin, 8 for cartpole_swingup, 4 otherwise |
| Batch size | 128 |
| Discount factor | 0.99 |
| Optimizer | Adam |
| Learning rate (actor, critic and encoder) | 1e-3 |
| SAC learning rate for $\alpha_{\text{SAC}}$ | 1e-4 |
| Discriminator learning rate (CMID only) | 1e-2 |
| SVEA coefficients | $\alpha_{\text{SVEA}} = 0.5$, $\beta_{\text{SVEA}} = 0.5$ |
| Target soft-update rate $\tau$ | critic 0.01, actor 0.05 |
| Actor update frequency | 2 |
| Actor log stddev bounds | $[-10, 2]$ |
| Latent representation dimension | 56 |
| Image size | $(84, 84, 3)$ |
| Image pad | 4 |
| Initial temperature | 0.1 |
| CMID loss coef $\alpha$ | 0.5 for cartpole_swingup, 0.1 otherwise |
| $k$ nearest neighbours | 5 |

**Table 2:** Hyperparameter values for both SVEA and SVEA-CMID.

**Encoder.** The encoder consists of 4 convolutional layers, each with a $3 \times 3$ kernel size and 32 channels. The first layer has a stride of 2, all other layers have a stride of 1. There is a ReLU activation between each of the convolutional layers. The convolutional layers are followed by a linear layer, normalisation, then a tanh activation. The encoder weights are shared between the actor $\pi$ and critic $Q$.

**Actor and critic.** Both the actor $\pi$ and critic $Q$ networks are MLPs consisting of two layers and a hidden dimension of 1024. There is a ReLU activation after each layer except the last layer.

**CMID discriminator.** The CMID discriminator is implemented as an MLP consisting of two layers and a hidden dimension of 1024. There is a ReLU activation after each layer except the last layer. The same conditional discriminator is used for all features in the representation so the inputs are one-hot encoded. This means the input size is: 56 (representation or permuted representation) + 56 (one-hot encoding of previous representation) + action size.

**Hyperparameters.** We tuned learning rate and CMID hyperparameters by grid search; other hyperparameters follow the original SVEA implementation. Table 2 shows the hyperparameters for all tasks.

**Hardware.** For each experiment run we use a single NVIDIA Volta V100 GPU with 32GB memory and a single CPU.

## C  Additional results

### C.1  Zero-shot generalisation

The zero-shot generalisation performance under correlation shift can be seen at the vertical dotted line in the graphs of Figure 4 and Figure 5. For completeness and to avoid loss of information caused by smoothing in the graphs, the numerical values of the zero-shot generalisation performance are provided in Table 3 and Table 4.

|            | cartpole_swingup    | walker_walk        | finger_spin        | hopper_stand       |
| ---------- | ------------------- | ------------------ | ------------------ | ------------------ |
| SVEA-CMID  | **746.0 ± 77.8**    | **793.5 ± 36.0**   | **939.5 ± 19.1**   | **826.0 ± 15.6**   |
| SVEA       | 233.1 ± 25.3        | 460.8 ± 50.7       | 633.8 ± 122.6      | 686.3 ± 170.8      |
| SVEA-TED   | 577.0 ± 152.0       | 542.7 ± 115.1      | 755.4 ± 75.8       | 623.5 ± 166.7      |
| CURL       | 262.4 ± 34.8        | 285.7 ± 54.3       | 386.3 ± 141.5      | 305.6 ± 160.4      |
| DrQ        | 201.2 ± 20.7        | 417.3 ± 32.1       | 843.3 ± 49.1       | 531.8 ± 182.6      |

**Table 3:** Zero-shot generalisation performance to *reversed correlation*. Returns are the average of 10 evaluation episodes over 5 seeds, showing ± standard error.

|            | cartpole_swingup    | walker_walk        | finger_spin        | hopper_stand       |
| ---------- | ------------------- | ------------------ | ------------------ | ------------------ |
| SVEA-CMID  | **878.8 ± 12.4**    | **815.3 ± 29.9**   | **953.2 ± 16.4**   | **816.1 ± 37.2**   |
| SVEA       | 371.5 ± 21.0        | 652.4 ± 34.3       | 680.1 ± 98.4       | 526.8 ± 182.2      |
| SVEA-TED   | 667.4 ± 120.6       | 560.7 ± 68.6       | 820.7 ± 59.6       | 643.3 ± 171.0      |
| CURL       | 523.8 ± 83.6        | 606.3 ± 50.8       | 561.7 ± 119.9      | 342.9 ± 151.1      |
| DrQ        | 521.6 ± 55.3        | 652.5 ± 26.4       | 872.0 ± 30.3       | 531.6 ± 149.5      |

**Table 4:** Zero-shot generalisation performance to *uncorrelated* variables. Returns are the average of 10 evaluation episodes over 5 seeds, showing ± standard error.

## C.2 Evaluation on each scenario

The results in Section 5 show the average returns over 10 evaluation episode for each seed, where a given scenario is selected based on the train/test probabilities depicted in Figure 3. To further assess performance, Figure 8 shows the average evaluation returns on 10 episodes for each object/colour combination on the cartpole swingup task with generalisation to reversed correlation. These results show that the correlation makes it difficult for SVEA to learn an optimal policy for any scenario, but with lower returns on the unlikely training scenarios in particular (cartpole A in green and cartpole B in blue). This explains the failure to generalise in Figure 4 when the correlation reverses, making the scenarios that were rare in training become frequent in testing at the vertical dotted line.

## C.3 Correlation strength.

The generalisation results in Section 5 show training with a 0.95 correlation (0.95 probability of being on the leading diagonal in Figure 3, and only 0.05 probability of being in the anti-diagonal scenarios). We conducted further analysis of different correlation strengths, denoting the sum of probabilities

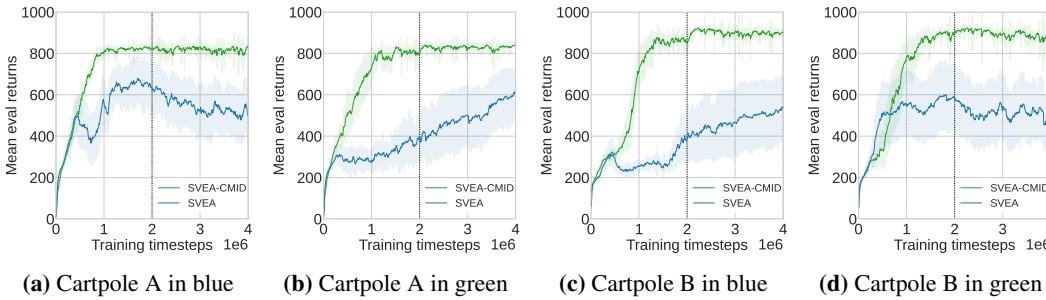

**(a)** Cartpole A in blue    **(b)** Cartpole A in green    **(c)** Cartpole B in blue    **(d)** Cartpole B in green

**Figure 8:** Evaluation of performance on each of the cartpole swingup scenarios for generalisation to reversed correlation, averaged over 10 evaluation episodes for 5 seeds.

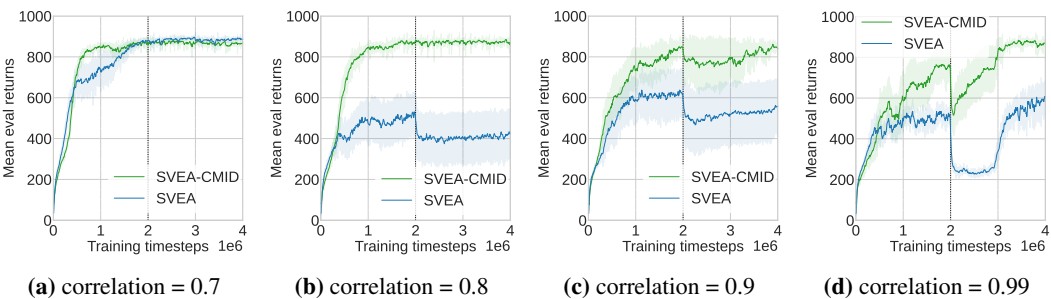

**(a)** correlation = 0.7     **(b)** correlation = 0.8     **(c)** correlation = 0.9     **(d)** correlation = 0.99

**Figure 9:** Generalisation to reversed correlation at the vertical dotted line with varying correlation strengths on the cartpole swingup task.

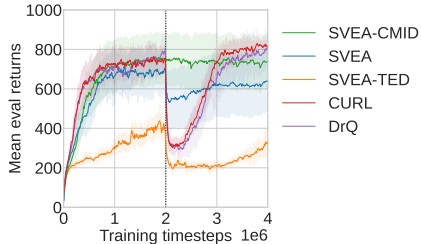

**Figure 10:** Generalisation to reversed correlation at the vertical dotted line on the cartpole swingup task with all image observations converted to greyscale.

on the leading diagonal as the correlation strength. The results for generalisation to the reversed correlation are shown in Figure 9. While the generalisation performance of SVEA decreases as the correlation gets stronger, SVEA-CMID consistently generalises well up to a very strong correlation of 0.99 at which point the performance deteriorates but still significantly improves the performance of SVEA in this setting.

### C.4 Greyscale images.

Our experiments use colour correlations to demonstrate the failure to generalise under correlation shifts. So we also demonstrate that the results still hold in greyscale images in Figure 10.

## D Environment variations

In Table 5, we provide a description of the differences between the two object variations (A and B) in each task, along with images of example observations for each object and colour combination. The exact specification of the world model for each task is available in our code.

| Environment | Variation | Blue | Green |
|---|---|---|---|
| cartpole_swingup | A - original cartpole | | |
| | B - wider cart, shorter pole | | |

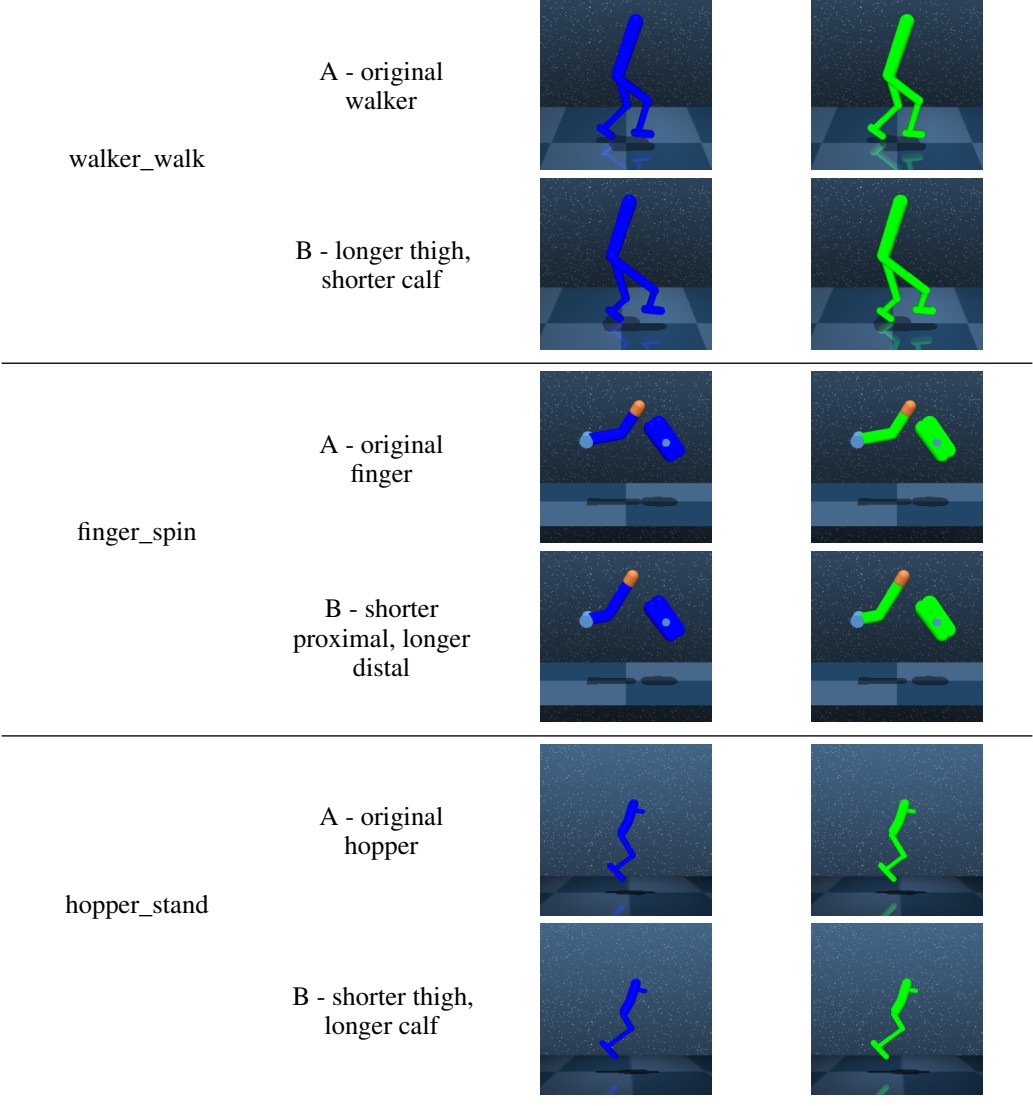

**Table 5:** Environment images

# E  Saliency maps

The full set of saliency maps, as described in Section 6, for each representation feature is provided in Figure 11 for a trained SVEA encoder and a trained SVEA-CMID encoder. The features are sorted in order of most active to least active based on the sum of attributions for each feature.

To create the saliency maps, we use the Captum open-source interpretability library for PyTorch (Kokhlikyan et al., 2020) to calculate the integrated gradients (Sundararajan et al., 2017) pixel attributions for each feature in the representation output of the encoder. We use an all black image as the baseline image for integrated gradients which is compared to the input image in Figure 7a. The absolute value of the attributions are overlaid onto the input image to create the saliency maps.

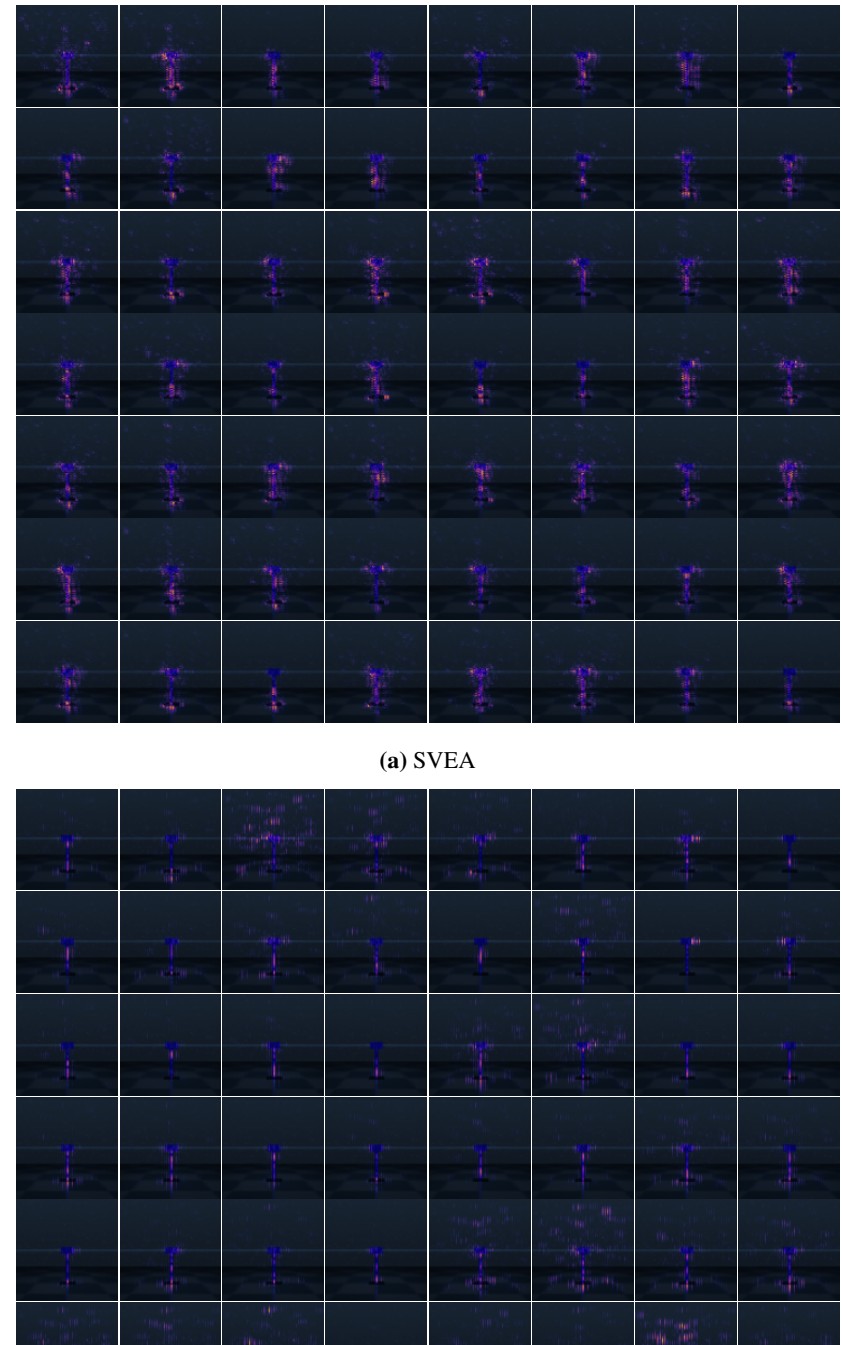

**(a)** SVEA

**(b)** SVEA-CMID

**Figure 11:** Saliency maps for each representation feature of a trained (a) SVEA and (b) SVEA-CMID encoder on the cartpole swingup task, sorted in order of highest total attributions to lowest.

